# Tree Prompting: Efficient Task Adaptation without Fine-Tuning

**John X. Morris**[*♣]   **Chandan Singh**[*◇]
**Alexander M. Rush**[♣]  **Jianfeng Gao**[◇]  **Yuntian Deng**[♠]

♣ Cornell University   ◇ Microsoft Research   ♠ Harvard University

{jxm3,arush}@cornell.edu,   {chansingh,jfgao}@microsoft.com,  dengyuntian@seas.harvard.edu

## Abstract

Prompting language models (LMs) is the main interface for applying them to new tasks. However, for smaller LMs, prompting provides low accuracy compared to gradient-based fine-tuning. Tree Prompting is an approach to prompting which builds a decision tree of prompts, linking multiple LM calls together to solve a task. At inference time, each call to the LM is determined by efficiently routing the outcome of the previous call using the tree. Experiments on classification datasets show that Tree Prompting improves accuracy over competing methods and is competitive with fine-tuning. We also show that variants of Tree Prompting allow inspection of a model's decision-making process.[1]

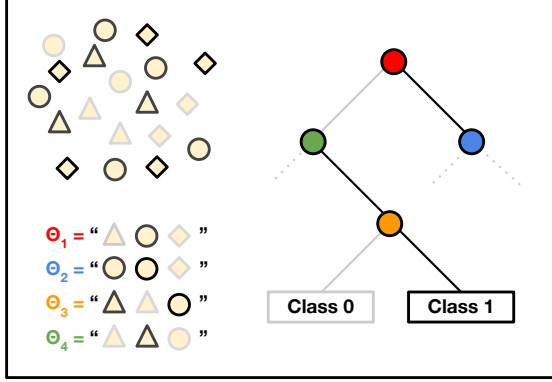

Figure 1: Illustration of Tree Prompting. At each node of the decision tree, a subset of training data is used to prompt the LM to partition the input space into sub-regions. This process is repeated until a classification decision is made at a leaf node.

## 1 Introduction

Pretrained language models (LMs) have made remarkable progress in recent years (Vaswani et al., 2017; Brown et al., 2020; OpenAI, 2023), but their large size makes them difficult to fine-tune with gradients for specific downstream tasks. As such, prompting has become the main interface for applying pretrained language models (LMs), where task-specific instructions are provided to guide an LM's behavior. The most common way to adapt LMs is to use few-shot in-context examples, where input-output pairs are shown to the model.

Yet, few-shot prompting has a clear downside. Prompt expressiveness is limited by the context length of the language model. This constraint prevents using more than a handful of examples for few-shot in-context learning, particularly in memory-constrained environments. If there is additional supervised data available for a task, users need to either ensemble together many prompts or back off to alternative LM fine-tuning approaches.

In this work, we propose Tree Prompting as an alternative method for incorporating task supervision. The key idea is to use training data to form a decision tree based on simple prompt-LM calls, with each prompt determined by the outcomes of previous calls. The method does not change the parameters of the language model, but instead uses its outputs to determine an effective tree structure. To determine the prompts used at each node of the decision tree, we propose a simple bagging-inspired approach that samples few-shot examples to find the most informative prompt (see Fig. 1). To convert LM outputs into split features for decision path determination, we consider both using a pre-defined verbalizer (Hu et al., 2022) and a more expressive $k$NN Prompting approach (Xu et al., 2023). To learn the tree structure, we employ a classic decision tree learning algorithm (Breiman et al., 1984). The constructed tree is a sparse representation of the fine-tuning data, incorporating a large number of few-shot examples, but only requiring a constant number of LM calls for inference.

Tree Prompting offers several advantages over existing prompting approaches. It allows users to

---

[1] *Equal contribution. Scikit-learn-compatible API for using Tree-Prompt is available at ⟳ github.com/csinva/tree-prompt.

easily incorporate large supervised training datasets without requiring larger contexts. It also allows experts to examine the decision-making process underlying a prediction in detail, which can be improved by combining with prompt generation methods. Finally, Tree Prompting can be adapted to be compatible with many existing LMs that are only accessible to the public via API calls. We demonstrate these advantages in experiments on multi-class classification benchmarks.

## 2   Background: Decision Trees

Decision trees are a classic model for classification and regression. They provide a graphical, intuitive model of decision-making, based on a cascading series of binary decisions[2]. At each node in the tree, a decision is made based on a single feature of the input, which leads to the next node, and ultimately to a leaf node representing a prediction.

**Learning**   Decision trees are constructed greedily in a top-down manner, starting from the root node, where all training data $(x, y)$ and features $\phi(x) \in \{0, 1\}^d$ are available. At each node, a feature that best splits the dataset into two subsets is chosen. The "best split" is determined by a criterion that measures the quality of a split. A commonly used criterion is the Gini impurity from the CART algorithm (Breiman et al., 1984). The selected feature creates two child nodes, each containing the subset of data that satisfies the respective split condition. This process is repeated recursively for each child node with the corresponding subset of the data until a stopping condition is met[3]. Each leaf node in the final decision tree represents a decision (such as a class label prediction), determined by the majority class of the instances in the leaf.

**Inference**   A decision tree makes predictions on unseen data by traversing the tree from the root node to a leaf node. Starting from the root, the feature value of the example corresponding to the split feature at the current node is used to determine whether the left child or the right child node is visited next. This process is repeated until a leaf node is reached. The class label associated with this leaf node is then used as the prediction.

---

[2]We exclusively focus on binary trees in this paper.

[3]Stopping conditions include reaching a maximum number of leaf nodes or when no feature improves the split quality.

## 3   Tree Prompting

Tree Prompting utilizes decision trees as a method of adapting LMs to specific tasks without fine-tuning the model. Assuming access to a set of text-label pairs $(x, y)$, the goal is to determine a tree to best classify this data. The algorithm then proceeds in a top-down manner, where at each node, it selects the best prompt based on the chosen method for finding prompt candidates and constructing split features.

However, unlike standard decision trees, we do *not* have access to a predetermined set of features $\phi(x)$. Tree Prompting instead constructs this feature function dynamically by constructing prompts. The value of a feature $\phi_i(x)$ is determined by running a prompt through the LM and mapping its response to a binary value.

A major benefit of utilizing decision trees in this setting is their efficiency at inference time. Constructing the tree lets us compactly represent a large amount of task-specific training examples. If each $\phi_i$ requires running one prompt, there are $2^D$ features. At inference time, we only need $D$ prompt calls to classify a single datapoint.

Our primary approach to find prompts for features $\phi_i(x)$ is to select random few-shot examples drawn from the training data, as shown in Figure 1. We take inspiration from bagging approaches (Breiman, 1996) that combine random training samples to produce complementary parallel models. By sampling random $x, y$ pairs from the task training data and passing them to an LM, we are effectively bagging small training splits. Each prompt is constructed by alternating classes with their corresponding labels in a templated form.

Once a prompt is constructed, the feature value is set using a pre-defined verbalizer to transform the LM's output (Hu et al., 2022). A verbalizer is a function that maps the LM's output probabilities into a discrete decision. A simple implementation of a verbalizer is to determine whether the predicted probability for the token *Yes/No* is higher. In this work, we experiment with two different verbalizers: the first maps the logits to class labels (such as *Positive/Negative* for binary sentiment classification), and the second more generic verbalizer maps the logits to *Yes/No*.

When the output logits of the LM are inaccessible[4], we can discretize the LM's outputs into

---

[4]Some recent LMs such as OpenAI's GPT-3.5 and GPT-4 do not provide output logits.

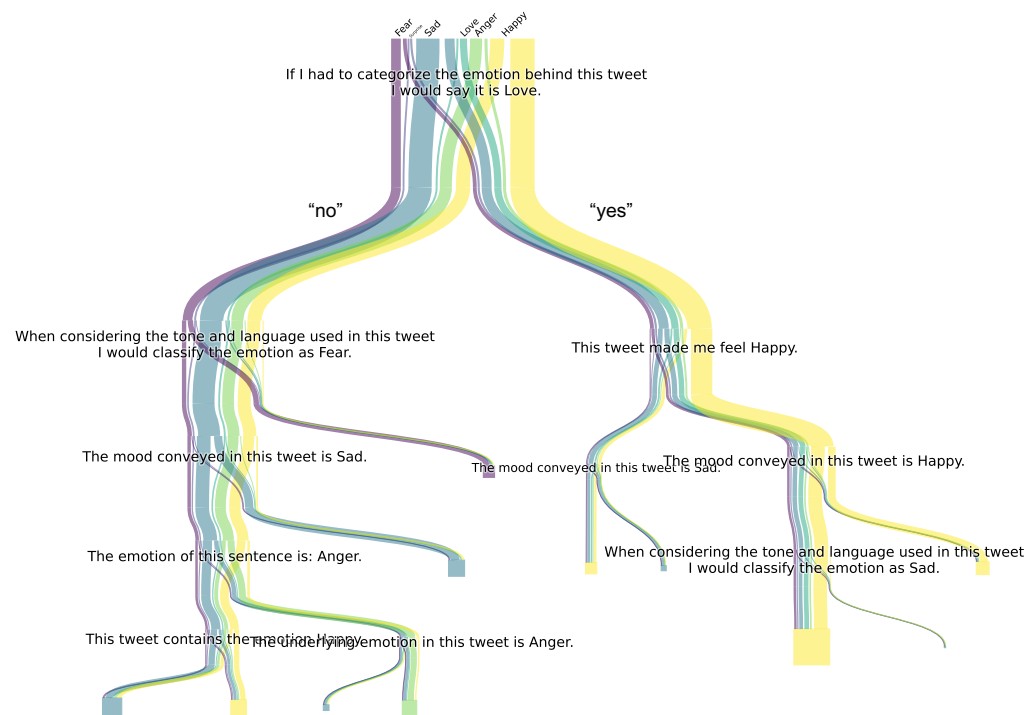

Figure 2: Extension to the Tree Prompting approach using instruction prompts on the Emotion dataset. Each path represents a decision sequence, and colors correspond to different emotions (classes). The line width indicates the number of instances within a particular class. As the decision process advances down the tree, the classes get separated.

categories defined by the verbalizer using word matching. With few-shot prompts, large LMs have empirically been found to respect the template format and output only labels that they have seen in the demonstrations most of the time (Brown et al., 2020).

### 3.1 Extensions

**Instruction Prompts** To leverage expert insights on specific tasks, we can instead use human-curated prompt candidates (Bach et al., 2022) as shown in Fig. 2. To diversify and enrich the pool of prompt candidates, we leverage the capabilities of GPT-3.5 to generate paraphrases of the original prompts. This method provides the ability to incorporate domain-specific knowledge, and the prompts are more interpretable compared to random few-shot examples. However, it might be less adaptable to novel or unique task specifications compared to the other automatic prompt candidate generation methods.

**Dynamic Prompts** Instead of pre-constructing random prompt-based features, we can generate dynamic prompts while building the decision tree. At each node, we conduct a discrete prompt search

to identify a list of prompt candidates that best explain the subset of data at this node. The prompt that best splits this subset into two further subsets is then selected. The prompt search algorithm used in this paper is iPrompt (Singh et al., 2023b), which employs an LM to generate potential prompt candidates, ranking them based on how well they explain the data. Dynamic prompts offer enhanced flexibility and adaptability, at the cost of additional computation.

$k$**NN Prompting Features** As a more expressive alternative to predefined verbalizers, we consider the $k$NN Prompting approach (Xu et al., 2023). $k$NN Prompting employs the label of the nearest neighbor in an anchor set as the split feature, with the distance measured in terms of the KL divergence between output probabilities. This approach allows for the use of a large number of examples at each node, extending beyond the limitations imposed by the restricted context window size, making it more expressive. A downside of this approach is its dependence on access to the LM's output logits. Moreover, as multiple prompts are utilized at each node of the decision tree, this can compromise the interpretability of the model.

| Approach | SST2 | SUBJ | MPQA | AGNews | CB | CR | DBPedia | MR | RTE | TREC | Emotion | FPB | IMDB | AVG | #LM Calls |
|---|---|---|---|---|---|---|---|---|---|---|---|---|---|---|---|
| Classes | 2 | 2 | 2 | 4 | 3 | 2 | 14 | 2 | 2 | 6 | 6 | 3 | 2 | - | - |
| **Finetuning** | | | | | | | | | | | | | | | |
| BERT[†] | 88.3 | 90.7 | 74.5 | 88.0 | 78.6 | 88.0 | 95.1 | 83.0 | 58.1 | 78.8 | 82.3 | - | - | - | 1.0 |
| GPT-2 Large[†] | 90.7 | 86.1 | 87.6 | 88.3 | 70.0 | 86.7 | 96.5 | 86.2 | 55.4 | 71.2 | 81.9 | - | - | - | 1.0 |
| **ICL GPT-2 Small** | | | | | | | | | | | | | | | |
| FSPrompting | 51.6 | 52.0 | 66.9 | 36.5 | 49.4 | 49.6 | 34.0 | 51.6 | 47.0 | 41.7 | 25.9 | 18.9 | 68.1 | 45.6 | 1.0 |
| Greedy | 48.4 | 49.3 | 73.0 | 73.2 | 51.2 | 50.4 | 71.7 | 64.7 | 54.6 | 51.4 | 37.4 | 61.8 | 63.9 | 57.8 | 40.0 |
| Boosting | 71.7 | 50.4 | 83.5 | 78.6 | 51.2 | 68.4 | 45.7 | 66.5 | 55.5 | 56.2 | 42.2 | 69.2 | 78.6 | 62.9 | 40.0 |
| TreePrompt | 72.3 | 50.4 | 83.5 | 80.6 | 55.4 | 68.0 | 82.9 | 66.7 | 56.0 | 63.4 | 43.0 | 67.9 | 77.5 | 66.7 | 8.8 |
| TreePrompt Ens | 72.7 | 50.4 | 83.3 | 81.2 | 59.5 | 68.2 | 87.6 | 66.5 | 55.9 | 71.7 | 43.3 | 68.7 | 78.7 | 68.3 | 32.6 |
| **ICL GPT-2 Medium** | | | | | | | | | | | | | | | |
| FSPrompting | 52.0 | 52.1 | 66.7 | 35.9 | 48.2 | 59.1 | 42.6 | 49.6 | 50.9 | 53.6 | 28.6 | 23.5 | 22.1 | 45.0 | 1.0 |
| Greedy | 83.1 | 74.3 | 63.8 | 70.8 | 64.3 | 50.4 | 78.9 | 77.9 | 54.8 | 63.0 | 46.8 | 61.8 | 74.3 | 66.5 | 40.0 |
| Boosting | 85.8 | 75.4 | 83.9 | 77.9 | 64.9 | 81.1 | 39.1 | 79.0 | 56.1 | 50.3 | 53.3 | 75.9 | 80.6 | 69.5 | 40.0 |
| TreePrompt | 83.6 | 76.2 | 83.9 | 78.3 | 66.7 | 80.6 | 90.5 | 78.8 | 54.9 | 72.8 | 53.3 | 78.4 | 81.1 | 75.3 | 6.2 |
| TreePrompt Ens | 85.5 | 75.9 | 85.2 | 79.8 | 64.9 | 80.6 | 94.5 | 79.0 | 55.6 | 77.9 | 54.9 | 79.4 | 81.8 | 76.5 | 35.5 |
| **ICL GPT-2 Large** | | | | | | | | | | | | | | | |
| FSPrompting | 81.6 | 52.2 | 50.0 | 49.5 | 40.5 | 64.6 | 33.6 | 59.4 | 51.0 | 54.7 | 27.5 | 52.1 | 50.8 | 51.4 | 1.0 |
| Greedy | 91.1 | 85.8 | 53.9 | 78.4 | 66.1 | 61.6 | 82.4 | 87.1 | 53.6 | 69.0 | 35.2 | 67.8 | 81.5 | 70.3 | 40.0 |
| Boosting | 91.9 | 87.4 | 84.0 | 79.9 | 65.5 | 85.4 | 25.0 | 86.5 | 54.0 | 59.8 | 45.0 | 78.6 | 90.6 | 71.8 | 40.0 |
| TreePrompt | 91.5 | 87.8 | 85.2 | 80.9 | 58.9 | 81.9 | 91.0 | 85.0 | 53.9 | 76.8 | 46.4 | 79.7 | 90.1 | 77.6 | 7.8 |
| TreePrompt Ens | 91.7 | 87.4 | 85.5 | 83.2 | 63.7 | 84.0 | 94.7 | 86.7 | 55.6 | 79.2 | 47.4 | 81.7 | 90.6 | 79.3 | 35.7 |
| **ICL GPT-2 XL** | | | | | | | | | | | | | | | |
| FSPrompting | 83.3 | 62.1 | 65.2 | 55.2 | 47.6 | 69.4 | 74.5 | 67.4 | 53.9 | 57.2 | 24.9 | 48.5 | 96.9 | 62.0 | 1.0 |
| Greedy | 90.5 | 77.9 | 76.3 | 83.7 | 62.5 | 73.6 | 88.8 | 84.2 | 57.7 | 63.5 | 35.2 | 65.8 | 86.3 | 72.8 | 40.0 |
| Boosting | 91.7 | 82.7 | 84.5 | 84.1 | 67.9 | 83.7 | 41.5 | 88.5 | 56.2 | 57.9 | 54.9 | 78.5 | 90.7 | 74.1 | 40.0 |
| TreePrompt | 83.6 | 76.2 | 83.9 | 78.3 | 66.7 | 80.6 | 90.5 | 78.8 | 54.9 | 72.8 | 53.3 | 78.4 | 81.1 | 75.3 | 6.6 |
| TreePrompt Ens | 85.5 | 75.9 | 85.2 | 79.8 | 64.9 | 80.6 | 94.5 | 79.0 | 55.6 | 77.9 | 54.9 | 79.4 | 81.8 | 76.5 | 36.2 |
| **ICL GPT-J** | | | | | | | | | | | | | | | |
| FSPrompting | 87.1 | 79.2 | 76.6 | 74.1 | 50.6 | 83.7 | 89.7 | 77.9 | 52.7 | 78.4 | 40.7 | 37.8 | 77.3 | 69.7 | 1.0 |
| Greedy | 91.3 | 89.8 | 85.4 | 83.9 | 72.0 | 87.2 | 93.8 | 89.6 | 58.9 | 81.0 | 49.4 | 69.4 | 93.7 | 80.4 | 40.0 |
| Boosting | 93.1 | 91.8 | 89.2 | 84.2 | 73.2 | 87.4 | 22.3 | 90.5 | 58.6 | 70.4 | 58.2 | 76.6 | 93.9 | 76.1 | 40.0 |
| TreePrompt | 91.5 | 92.2 | 87.4 | 85.5 | 73.2 | 87.8 | 97.5 | 90.5 | 59.8 | 83.2 | 59.6 | 76.5 | 93.1 | 82.9 | 8.9 |
| TreePrompt Ens | 93.1 | 92.2 | 88.5 | 85.8 | 75.0 | 87.5 | 98.2 | 90.5 | 59.2 | 87.2 | 61.9 | 79.2 | 93.7 | 84.0 | 32.3 |
| **ICL LLAMA-2 7B** | | | | | | | | | | | | | | | |
| FSPrompting | 92.7 | 55.1 | 78.6 | 84.9 | 55.4 | 90.9 | 94.0 | 90.8 | 59.2 | 80.2 | 37.5 | 70.2 | 60.1 | 73.1 | 1.0 |
| Greedy | 95.1 | 84.1 | 85.3 | 86.6 | 64.9 | 90.4 | 98.2 | 94.7 | 71.7 | 83.9 | 47.6 | 70.2 | 86.0 | 81.4 | 40.0 |
| Boosting | 94.4 | 89.2 | 86.1 | 87.2 | 76.2 | 90.9 | 21.5 | 92.7 | 74.0 | 39.3 | 55.1 | 81.6 | 91.9 | 75.4 | 40.0 |
| TreePrompt | 93.6 | 89.2 | 85.2 | 88.5 | 75.0 | 88.5 | 98.6 | 93.9 | 69.7 | 88.0 | 54.8 | 84.0 | 93.8 | 84.8 | 6.6 |
| TreePrompt Ens | 94.5 | 90.0 | 86.6 | 88.2 | 76.8 | 89.2 | 98.6 | 93.8 | 73.6 | 89.7 | 57.2 | 84.9 | 93.5 | 85.9 | 35.4 |

Table 1: Main results. ICL: In Context Learning. ICL Prompting and ICL Prompting Ensemble use 128 examples per class to construct the prompt. [†]: results taken from Xu et al. (2023).

**Tree Ensembles** Trees generated via Tree Prompting can be used to construct typical tree ensembles such as random forests (Breiman et al., 1984) or gradient-boosted trees (Freund et al., 1996) by using a Tree Prompting tree as the base estimator. This incurs very little overhead when using a fixed set of prompts, as the split features can be shared across all trees after being computed once.

## 4 Experimental Setup

**Datasets** We evaluate Tree Prompting on 13 text classification datasets. Among them are binary classification datasets SST2 (Socher et al., 2013), SUBJ (Pang and Lee, 2004; Wiebe et al., 2005), MPQA (Deng and Wiebe, 2015), CR (Hu and Liu, 2004), MR (Pang and Lee, 2005), RTE (Dagan et al., 2006), IMDB (Maas et al., 2011), and multi-class classification datasets AGNews (Zhang et al., 2015), CB (De Marneffe et al., 2019), DB-Pedia (Zhang et al., 2015; Lehmann et al., 2015), TREC (Li and Roth, 2002; Hovy et al., 2001), FPB (Malo et al., 2014), and Emotion (Saravia et al., 2018). Appendix A.1 provides dataset statistics in Table 6, and examples in Table 7.

**Model Settings** For the LM, we run experiments using five pretrained models: GPT-2 Small (117M parameters), GPT-2 Medium (355M parameters), GPT-2 Large (774M parameters), GPT-2 XL (1.5B parameters) (Radford et al., 2019), and GPT-J (Wang and Komatsuzaki, 2021) (6B parameters).

**Baselines** We compare our approach Tree Prompting, *TreePrompt*, to standard fine-tuning. In addition, we also compare against a conventional prompting baseline *FSPrompting*, which directly uses few-shot example demonstrations as the prompt. We also compare the performance of

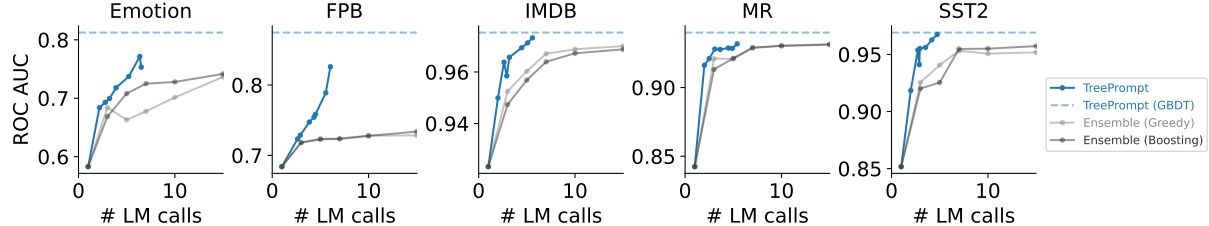

Figure 3: Performance as a function of the number of LM evaluations per example (#LM calls). We use GPT-J as the base LM and class names as verbalizers. GBDT, gradient-boosting tree using Tree Prompting as the base classifier, fitted to a maximum of 40 LM calls provides an upper bound of accuracy we can get on individual datasets.

our ensembling approach, *TreePrompt Ens*[5], with two baseline ensembling strategies: *Greedy*, which adds prompts to an ensemble in order of their cross-validation accuracy), and *Boosting*, which adds prompts to an ensemble using AdaBoost (Freund and Schapire, 1997; Hou et al., 2022; Pitis et al., 2023).

## 5 Results

### 5.1 Classification Accuracy

Our main results are summarized in Table 1. The table compares Tree Prompting across multiple language model sizes to other few-shot prompting and ensembling approaches, as well as to gradient-based fine-tuning. Approaches are allotted a maximum of 40 LM inference calls per example.

Results show that Tree Prompting outperforms basic few-shot prompting and also ensembling-based approaches across model sizes and almost all datasets. The performance difference is particularly large across smaller model classes. For instance, while FSPrompting averages an accuracy of 44.3% with GPT-2 Small, Tree Prompting elevates this to 60.5%. Tree Prompting can also be ensembled, which produces accuracy improvements at the cost of more LM calls.

We also compare Tree Prompting to gradient based fine-tuning, particularly on GPT-2 Large. Results show that Tree Prompting is less stable than fine-tuning, performing poorly on some tasks, but outperforms it on 5 of 10 tasks. This result shows that Tree Prompting can learn well from task supervision at the cost of additional runtime queries. (Tree Prompting could likely perform better compared to fine-tuning if we increased the maximum number of prompts beyond 40.)

Relative to the baselines, Tree Prompting is typically able to outperform them all while making fewer queries than Greedy and Boosting. Tree Prompting makes large improvements over few-show prompting in most cases, even when the model size is large. We observe a failure of the boosting strategy when the number of classes is large (specifically for 14-class DBPedia). Tree Prompting with gradient boosting generally gives an increase at performance at the cost of a 5.7-times increase in queries.

### 5.2 Inference Efficiency

As computing outputs from language models can be costly, particularly with large LMs, the efficiency of Tree Prompting at inference time is crucial. In Fig. 3, we plot test accuracy against the number of language model evaluations per example (#LM Calls) to gauge this efficiency[6]. Tree Prompting frequently surpasses competing ensemble strategies in performance under the same number of LM calls, indicating an improvement in efficiency. This gain is more significant for multiclass datasets, such as Emotion and Financial phrasebank (FPB).

To establish an upper bound of accuracy, we consider Tree Prompting ensembling. This approach generally achieves the best test performance across all methods, although it also demands more LM calls than a single tree (up to 40 calls).

### 5.3 Interpretability and Dynamic Prompts

A practical benefit of decision trees is increased interpretability. Each node of the decision tree can be inspected, offering insights into the decision-making process when predicting the label of a given input. Our few-shot approach for Tree Prompting

---

[5]This ensemble uses scikit-learn's GradientBoostingClassifier with its default parameters: 100 trees, each with max depth 3, with a learning rate of 0.1.

[6]The mean number of LM calls may differ from the max depth of a tree, as reaching a leaf node can require fewer calls in an unbalanced tree.

| | CB | CR | SUBJ | MPQA | RTE | TREC | MR | DBPedia | SST2 | AGNews | AVG |
|---|---|---|---|---|---|---|---|---|---|---|---|
| Train Size (k) | 0.25 | 1.77 | 2.49 | 5.45 | 8.00 | 8.60 | 8.66 | 50.00 | 67.35 | 120.00 | - |
| $k$NN Prompting[†] | **62.1** | **87.5** | 54.1 | **86.7** | 87.6 | **84.8** | 83.6 | 96.7 | 85.5 | 87.6 | 81.6 |
| Tree Prompting | 58.2 | 86.6 | **54.4** | 85.9 | **88.4** | **84.8** | **86.2** | **97.2** | **90.3** | **88.8** | **82.1** |

Table 2: Comparison between Tree Prompting and $k$NN Prompting. Both approaches use GPT-2 Large as the base LM. Tree Prompting uses predictions from $k$NN Prompting to construct split features. Tree Prompting results are averaged over 5 random seeds. [†]: results taken from Xu et al. (2023).

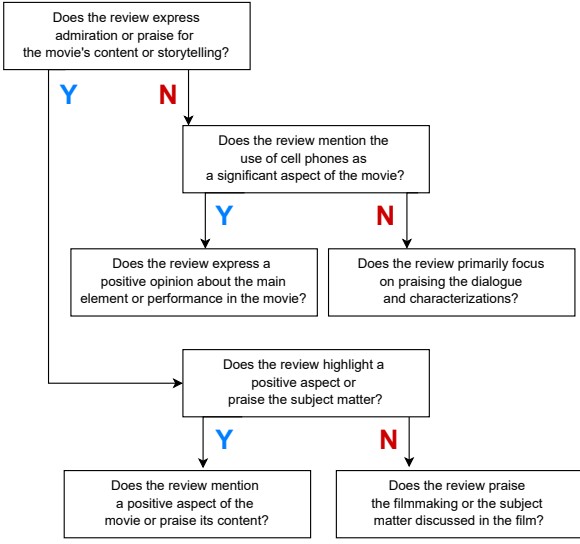

Figure 4: Tree Prompting tree learned using dynamic prompts on the MR dataset. We use GPT-4 for prompt generation in AutoPrompting and GPT-J as the base LM in Tree Prompting.

is challenging to interpret, but we can instead use interpretable prompts that are human-curated or dynamically constructed.

Fig. 2 demonstrates an instance of a decision tree learned from human-curated prompts on the Emotion dataset, where different colors represent the true labels of the data. At the root node, a high-level query is posed regarding whether the tweet's underlying emotion is *love*. Deeper in the tree, more granular questions are presented, e.g. whether the sentiment of the sentence is *anger*.

Dynamic prompts offer an additional advantage over human-curated prompts; they are capable of better reflecting the specific subset of data at each node, making the decision process more aligned with the data distribution at each tree node. Fig. 4 shows a tree learned using iPrompt to create dynamic prompts at each node of the tree. Prompts are suggested by GPT-4 and reranked according to the iPrompt algorithm with the verbalizer *Yes/No* corresponding to the positive/negative classes of

the MR dataset.

## 5.4 Comparison with $k$NN Prompting

Nonparametric methods like $k$NN Prompting (Xu et al., 2023) can be employed to improve model expressivity, which allows using multiple prompts per node and avoids the reliance on pre-defined verbalizers. Table 2 provides a comparison between Tree Prompting and $k$NN Prompting. In this comparison, Tree Prompting uses $k$NN Prompting predictions as split features[7]. The results show that Tree Prompting outperforms vanilla $k$NN Prompting on most datasets, potentially due to its added flexibility of partitioning the input space using the decision tree, although it underperforms on three of the smaller datasets CB (250 training examples), CR (1.77k training examples), and TREC (5.44k training examples).

## 5.5 Comparison to Larger LMs

Tree Prompting allows enhancing the performance of small LMs to match the performance of large LMs, as shown in Table 3. For these experiments instead of few-shot prompting we use instruction prompts curated from PromptSource (Bach et al., 2022)[8]. In this setting, even GPT-2 Small paired with Tree Prompting Ensemble is competitive against GPT-3 (text-davinci-003), outperforming it on two datasets (FPB and MR), albeit being slightly worse on the other two datasets (IMDB and SST2). With the larger LM GPT-J, Tree Prompting outperforms GPT-3 with conventional prompting across all datasets, demonstrating the potential of using a smaller model in a decision tree repeatedly to outperform a larger model, which might be useful in resource-constrained scenarios.

---

[7]We binarize $k$NN Prompting predictions, which are multi-class labels, into multiple split features (each evaluating whether the output matches a certain class).

[8]Initial experiments showed that instruction prompts outperform few-shot prompts for GPT-3.

| LM | Approach | FPB | MR | IMDB | SST2 | AVG |
|---|---|---|---|---|---|---|
| GPT-3 | Zero-Shot Instruction Prompting | 39.6 | 82.7 | 75.6 | 90.5 | 72.1 |
| | AutoPrompting (Singh et al., 2023a) | 57.2 | 77.4 | 86.6 | 82.4 | 75.9 |
| GPT-2 Small | TreePrompt Ensemble | 71.4 | 77.5 | 85.8 | 80.8 | 78.9 |
| **GPT-J** | **TreePrompt Ensemble** | **80.2** | **91.3** | **94.5** | **93.7** | **89.9** |

Table 3: Tree Prompting with supervision achieves comparable accuracy to GPT-3 zero-shot and supervised auto-prompting. Tree Prompting uses instruction prompts, class names as the verbalizer, and fits gradient-boosted trees with up to 40 prompts. Averaged over 3 random seeds.

| Verbalizer | FPB | MR | IMDB | SST2 | AVG |
|---|---|---|---|---|---|
| **Yes/No** | | | | | |
| Greedy | 58.9 | 72.7 | 58.8 | 75.7 | 66.5 |
| Boosting | 57.8 | 71.6 | 58.7 | 75.8 | 66.0 |
| TreePrompt | **64.4** | **79.4** | **68.6** | **77.2** | **72.4** |
| **Class Names** | | | | | |
| Greedy | 59.6 | 64.5 | 62.3 | 78.2 | 66.1 |
| Boosting | 61.9 | 65.2 | 62.4 | 77.1 | 66.6 |
| TreePrompt | **74.2** | **73.4** | **65.7** | **80.4** | **73.4** |

Table 4: Accuracy with different verbalizers. We employ GPT-2 Small as the LM, limiting to a maximum of 5 average calls during inference.

| Prompt Source | FPB | MR | IMDB | SST2 | AVG |
|---|---|---|---|---|---|
| **Few-shot** | | | | | |
| Greedy | 59.6 | 64.5 | 62.3 | 78.2 | 66.1 |
| Boosting | 61.9 | 65.2 | 62.4 | 77.1 | 66.6 |
| TreePrompt | **74.2** | **73.4** | **65.7** | **80.4** | **73.4** |
| **Instructions** | | | | | |
| Greedy | 71.8 | 83.3 | 88.2 | 86.5 | 82.4 |
| Boosting | 77.4 | 84.2 | 91.3 | 87.6 | 85.1 |
| TreePrompt | **80.9** | **85.1** | **92.0** | **88.5** | **86.6** |

Table 5: Comparative results using different prompt sources. We use GPT-2 Small with class names as the verbalizer, limiting the LM to a maximum of 5 average calls during inference.

## 6 Analysis

### 6.1 Verbalizer Sensitivity

Table 4 shows the robustness of different approaches when employing a generic *Yes/No* verbalizer versus a class-name verbalizer. The results show that Tree Prompting consistently outperforms the baseline regardless of the verbalizer used, delivering decent performance even when using the generic *Yes/No* verbalizer. This feature could be useful in applications where class names are not meaningful words, such as in distinguishing between texts generated by different decoding settings (Naseh et al., 2023). Table 8 in Appendix A.3 shows full performance sensitivity results across different settings for the underlying LM, verbalizer,

and source of prompts.

### 6.2 Prompt Source Sensitivity

Table 5 examines the sensitivity of various approaches to the source of prompt candidates. The comparison between using instruction prompts and few-shot prompts demonstrates that Tree Prompting consistently outperforms baselines regardless of the source of prompt candidates. It's worth noting that instruction prompts generally result in better performance than few-shot prompts, corroborating previous findings that in-context learning with a single prompt can work as well as multiple data demonstrations (Le Scao and Rush, 2021). However, curating instruction prompts requires extra human effort, since new prompts must be written for each new dataset.

### 6.3 Sample Complexity

Fig. 5 visualizes the performance of Tree Prompting in relation to the fraction of training samples used for training. When compared to baseline ensembling techniques, Tree Prompting sometimes underperforms in low-data regimes (on FPB, IMDB, and MR), but it eventually outperforms baselines as more training data is available.

## 7 Related Work

**Prompting Language Models** The rise of large language models (LMs) has led to a surge in the development of effective prompting methods (Strobelt et al., 2022; Lu et al., 2022; Bach et al., 2022; Logan IV et al., 2022; Zhong et al., 2022; Singh et al., 2023b). Building on top of these methods, emsembling techniques for averaging multiple LM calls have shown that they often improve performance (Jiang et al., 2020; Zhang et al., 2023a), e.g. boosting (Hou et al., 2022; Pitis et al., 2023). Chain prompting (Wang et al., 2022; Press et al., 2022; Chase, 2023; Rush, 2023) is a widely used method that divides complex tasks into manageable subtasks, linking these via prompt-LM calls.

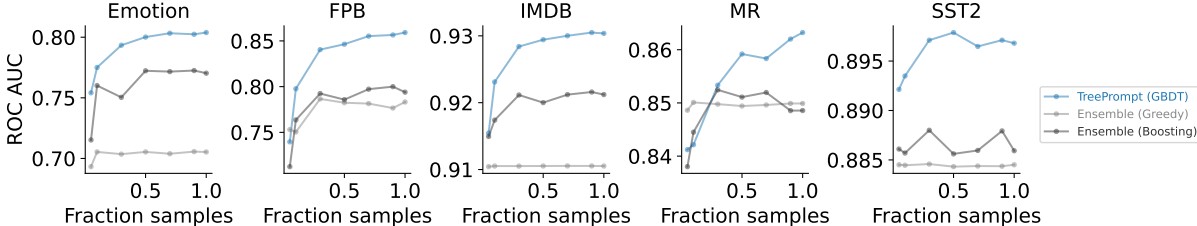

Figure 5: Accuracy plotted against the fraction of samples used for training. The performance improvement facilitated by Tree Prompting (shown here with gradient-boosting) becomes more noticeable as the number of training samples escalates. We use GPT-2 with *Instruction* prompts and a set of 10 prompts for this visualization.

This approach has proven effective across various applications, aligning with our intuition underlying this work: an LM can handle individual steps of a task more accurately than executing the task in full (Ma et al., 2023; Madaan et al., 2023; Zhang et al., 2023b). However, while chain prompting links prompt-LM calls, our approach organizes them within a decision tree, learning the tree structure and selecting appropriate prompts for each node.

Frugal GPT (Chen et al., 2023) also bears relevance to our work, proposing a cascade of LMs that stops when an intermediate output is considered reliable, resulting in better computational efficiency. Viewed from the perspective of decision trees, this approach resembles a right-branching decision tree.

Concurrent to our work, Tree of Thoughts (Yao et al., 2023; Long, 2023) organizes LM-generated "thoughts" within a tree structure for solution search. While we also use a tree structure, our aim is to partition the input space to simplify the LM's tasks at lower tree levels. We search the tree's structure and the prompt at each node during training, while keeping these elements static during inference. In contrast, Tree of Thoughts adjusts node prompts dynamically based on upper-level results. This sets it apart from our approach, where prompts remain constant post-training. Collectively, these works demonstrate the growing interest in merging tree structures with LMs for task decomposition, albeit with varied focuses and methodologies.

**Decision Tree Applications** Dating back decades, decision trees have been a prevalent choice in the realms of classification and regression problems (Costa and Pedreira, 2022). In the field of natural language processing, decision trees and their ensemble variants such as Random Forest (Breiman, 2001), Gradient-boosted Trees (Freund et al., 1996), XGBoost (Chen and Guestrin, 2016), and BART (Chipman et al., 2010) have found use in areas like part-of-speech tagging (Magerman, 1995), syntactic parsing (Collins, 1997), and text classification (Sebastiani, 2002; Singh et al., 2023a). However, these studies predominantly utilize pre-defined textual features within their decision tree frameworks, contrasting our approach where the decision tree is used to direct the language model's behavior.

**Decision Trees for Interpretability** Decision trees have also been applied to increase the interpretability of neural models. For example, Wan et al. (2021) used a decision tree structure where each node is a neural classifier for image classification. Zhang and Zhu (2019) learned a decision tree to explain the decisions made by an image classifier post hoc. While these works primarily target vision-based applications, we adopt a similar strategy for natural language processing, where each node in our decision tree embodies a distinct prompt-LM call. Furthermore, our dynamic prompt setting enables the concurrent learning of prompts and the decision tree structure, distinguishing our method from conventional decision tree applications that function within a pre-defined feature space.

## 8  Conclusions and Future Work

We introduce the Tree Prompting approach, a use of decision trees for task adaptation. Experiments demonstrate that Tree Prompting can offer improved performance across various text classification tasks while still remaining efficient during inference. On many tasks, the model is competitive with gradient fine-tuning. Additionally, the approach can be used with dynamic prompt creation to yield interpretable models.

Our results suggest a future direction of exploring a flexible and modularized assembly of models. One exciting direction is to extend Tree Prompting

to generalize to tasks beyond text classification, using previous outputs to guide subsequent prompts and LMs. Further exploration could involve extending Tree Prompting to jump across nodes in the tree (similar to Long (2023)) or introduce cycles in the tree (similar to Besta et al. (2023)), and ultimately developing a program of prompts by navigating various nodes in a decision tree as though calling different functions. Another direction could explore incorporating different criteria into the tree-building algorithm, e.g. fairness (Jo et al., 2022), sparsity (Hu et al., 2019; Tan et al., 2022), or smoothness (Agarwal et al., 2022).

## 9 Limitations

**Sample Complexity** While Tree Prompting's adaptability and flexibility are its strengths, they also contribute to its higher sample complexity. As shown in Sec. 6.3, Tree Prompting lags behind few-shot prompting in low-data environments. Decision trees inherently risk overfitting, particularly when dealing with numerous features. This shortcoming can be partially offset through the use of larger training sets, and by restricting the tree's size in relation to the training set size.

**Training Cost** Although Tree Prompting demands fewer LM calls during inference compared to analogous techniques, its training process, which involves learning the decision tree, requires computing prompt features for every example in the associated data subset at each node. This can be resource-intensive for large LMs. Additionally, when paired with dynamic prompts that leverage automatic prompting methods (which are typically computation-heavy), the training process can be substantially expensive as each node necessitates running the autoprompting method once.

**Interpretability** While decision trees are typically celebrated for their interpretability, the interpretability of Tree Prompting is bounded by the nature of the prompts and the verbalizer. Specifically, when employing a pre-defined prompt, its interpretability may not be as intuitive as that of dynamic prompts. If the prompt itself (such as when using few-shot demonstrations) lacks interpretability, the entire decision tree's interpretability is likely to be compromised.

## 10 Acknowledgements

JXM is supported by an NSF GRFP. YD is supported by an Nvidia Fellowship and NSF 2242302. AMR is supported by NSF 2242302, NSF CAREER 2037519, and a Sloan Fellowship. We would also like to thank Harvard University FAS Research Computing for providing computational resources.

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

# A  Appendix

## A.1  Data

Table 6 presents dataset statistics, and Table 7 shows example input, output pairs from each dataset.

## A.2  Tree Visualizations

Appendix A.2 shows another example tree learned on the MR dataset.

## A.3  Full Ablation Results

Full ablation results for different choices of prompts and verbalizers can be found in Table 8.

| | CB | CR | RTE | TREC | Emotion | FPB | SST2 | SUBJ | MPQA | MR | AGNews | DBPedia | IMDB |
|---|---|---|---|---|---|---|---|---|---|---|---|---|---|
| Classes | 3 | 2 | 2 | 6 | 6 | 3 | 2 | 2 | 2 | 2 | 4 | 14 | 2 |
| Train Size (k) | 0.25 | 1.77 | 2.49 | 5.45 | 5.45 | 2.31 | 67.35 | 8.00 | 8.60 | 8.66 | 120.00 | 50.00 | 25.00 |

Table 6: Dataset statistics

| Dataset | Text | Label |
|---|---|---|
| SST2 | that loves its characters and communicates something rather beautiful about human nature | positive |
| SUBJ | the script isn't very good ; not even someone as gifted as hoffman ( the actor ) can make it work . | subjective |
| MPQA | victory of democracy | positive |
| AGNews | Wall St. Bears Claw Back Into the Black (Reuters). "Reuters - Short-sellers, Wall Street's dwindling band of ultra-cynics, are seeing green again." | business |
| CB | Premise: "Do you mind if I use your phone?" Ronni could see that Guido's brain was whirring. Hypothesis: Guido's brain was whirring | entailment |
| CR | i didn 't have any major problems installing this software . | positive |
| DBPedia | Geoffrey D. Falksen (born July 31 1982) is an American steampunk writer. | artist |
| MR | the film is flat . | negative |
| RTE | Sentence 1: No Weapons of Mass Destruction Found in Iraq Yet.  Sentence 2: "Weapons of Mass Destruction Found in Iraq. | not_entailment |
| TREC | What 's known as The queen of Drinks ? | entity |
| FPB | According to Gran , the company has no plans to move all production to Russia , although that is where the company is growing . | neutral |
| IMDB | would put this at the top of my list of films in the category of unwatchable trash! [...] | negative |
| Emotion | i can go from feeling so hopeless to so damned hopeful just from being around someone who cares and is awake | sadness |

Table 7: Data examples

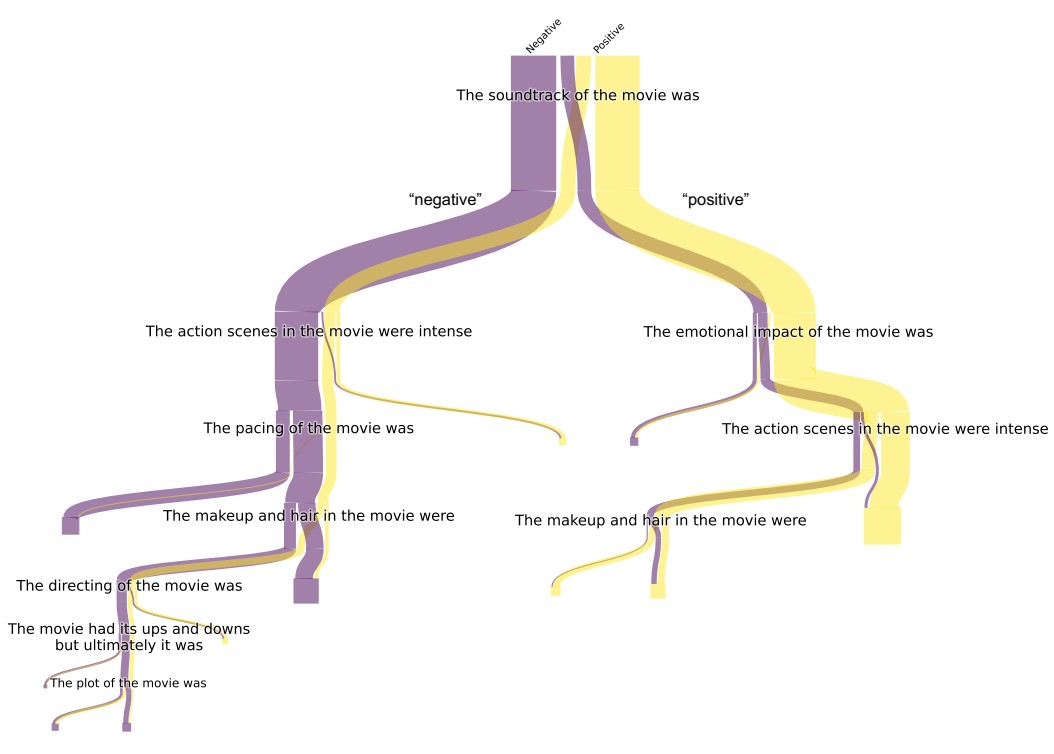

Figure 6: Example tree for the MR dataset. We use GPT-J and search for 10 instruction prompts.Class names (positive/negative) are used as the verbalizer.

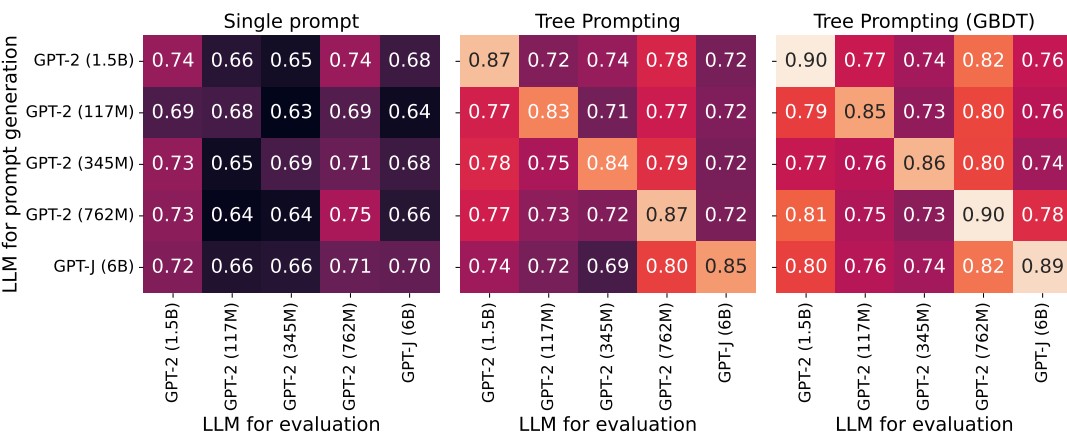

Figure 7: Performance when varying the LLM used to select prompts versus evaluate performance. Each heatmap shows the test ROC AUC achieved averaged across 4 datasets (SST2, Emotion, Financial Phrasebank, and Rotten tomatoes) using either a single prompt, Tree Prompting, or a Tree Promptinggradient-boosting ensemble. Unsurprisingly, performance is best when building a tree of prompts on the same dataset as is used for testing. Trees from Tree Prompting still induce better performance when transferring to new LLMs than individual prompts. All prompting strategies select from 40 "Instructions" prompts as in Table 5).

| Dataset | Model | Prompts | Verbalizer | Ensemble (Greedy) | Ensemble (Boosting) | **Tree Prompting** |
|---|---|---|---|---|---|---|
| Emotion | GPT-2 | 1-Shot | Class names | 0.56 | 0.56 | **0.59** |
| | | 5-Shot | Class names | 0.59 | 0.54 | **0.63** |
| | | Instruction | Class names | 0.76 | 0.63 | **0.77** |
| | GPT-J | 1-Shot | Class names | 0.71 | 0.68 | **0.72** |
| | | 5-Shot | Class names | 0.72 | 0.70 | **0.76** |
| | | Instruction | Class names | 0.71 | 0.57 | **0.73** |
| FPB | GPT-2 | 1-Shot | Class names | 0.62 | 0.60 | **0.74** |
| | | | Yes/no | 0.58 | 0.59 | **0.64** |
| | | 5-Shot | Class names | 0.73 | 0.74 | **0.74** |
| | | | Yes/no | 0.72 | 0.73 | **0.76** |
| | | Instruction | Class names | 0.77 | 0.72 | **0.81** |
| | GPT-J | 1-Shot | Class names | 0.72 | 0.72 | **0.76** |
| | | | Yes/no | 0.57 | 0.57 | **0.72** |
| | | 5-Shot | Class names | 0.72 | 0.72 | **0.84** |
| | | | Yes/no | 0.58 | 0.58 | **0.65** |
| | | Instruction | Class names | 0.80 | 0.77 | **0.86** |
| IMDB | GPT-2 | 1-Shot | Class names | 0.62 | 0.62 | **0.66** |
| | | | Yes/no | 0.59 | 0.59 | **0.69** |
| | | Instruction | Class names | 0.91 | 0.88 | **0.92** |
| | GPT-J | 1-Shot | Class names | 0.96 | 0.96 | **0.97** |
| | | | Yes/no | 0.95 | 0.96 | **0.97** |
| | | Instruction | Class names | 0.93 | 0.92 | **0.95** |
| MR | GPT-2 | 1-Shot | Class names | 0.65 | 0.64 | **0.73** |
| | | | Yes/no | 0.72 | 0.73 | **0.79** |
| | | 5-Shot | Class names | 0.50 | 0.50 | **0.51** |
| | | | Yes/no | 0.58 | 0.58 | **0.64** |
| | | Human-1 | Class names | 0.84 | 0.83 | **0.85** |
| | GPT-J | 1-Shot | Class names | 0.92 | 0.92 | **0.93** |
| | | | Yes/no | 0.88 | 0.87 | **0.93** |
| | | 5-Shot | Class names | 0.93 | 0.95 | **0.96** |
| | | | Yes/no | 0.80 | 0.80 | **0.93** |
| | | Human-1 | Class names | 0.86 | 0.85 | **0.88** |
| SST2 | GPT-2 | 1-Shot | Class names | 0.77 | 0.78 | **0.80** |
| | | | Yes/no | 0.76 | 0.76 | **0.77** |
| | | 5-Shot | Class names | 0.61 | 0.61 | **0.74** |
| | | | Yes/no | 0.75 | 0.75 | **0.80** |
| | | Instruction | Class names | 0.88 | 0.87 | **0.88** |
| | GPT-J | 1-Shot | Class names | 0.93 | 0.94 | **0.97** |
| | | | Yes/no | 0.90 | 0.90 | **0.94** |
| | | 5-Shot | Class names | 0.93 | 0.94 | **0.97** |
| | | | Yes/no | 0.58 | 0.58 | **0.58** |
| | | Instruction | Class names | 0.87 | 0.87 | **0.90** |

Table 8: Performance (ROC AUC) for different ensembling strategies when using at most 5 LM calls across different datasets, models, prompts, and verbalizers. *Emotion* is not compatible with a *Yes/No* verbalizer, so it has two fewer rows. Some rows are missing for datasets for which 5 demonstrations are too long to fit in context.