# OpenReview forum: "Tree Prompting: Efficient Task Adaptation without Fine-Tuning"
_EMNLP/2023/Conference — EMNLP 2023 Main_

### Official Review · Reviewer_VnSE · 2023-08-02

**Typos Grammar Style And Presentation Improvements:** 1. In Table 2, does Tree Prompting ac…
**Soundness:** 4

**Excitement:**

4: Strong: This paper deepens the understanding of some phenomenon or lowers the barriers to an existing research direction.

**Paper Topic And Main Contributions:**

The paper presents an interesting approach to address the limitations of few-shot in-context examples in language models. These in-context examples have shown effectiveness, but their expressiveness is limited by the context length. To tackle this problem, the authors propose a solution called Tree Promptings, which involves using training data to create a decision tree based on prompts. This decision tree is constructed using a bagging-inspired approach. Additionally, the paper utilizes pre-defined verbalizers and a kNN Prompting approach to split features.

The experimental results demonstrate that the proposed method outperforms various prompting baselines on different text classification tasks. Notably, the approach remains efficient during inference, making it a promising solution for improving performance in a practical and scalable manner.

**Questions For The Authors:**

1. Has the author explored training Tree Prompt on smaller language models, such as GPT-2 small, and subsequently evaluating its performance on larger language models? Additionally, did the author manually inspect the trained Tree Prompt and subsequently test it on large language models?

**Reasons To Accept:**

1. The proposed Tree Prompt demonstrates enhanced performance on diverse classification datasets, while maintaining efficiency during inference.
2. The paper conducts comprehensive experiments to showcase the effectiveness and robustness of the proposed method.
3. The paper presents a clear and promising direction for future prompting design.
4. The code provided in the paper ensures reproducibility.

**Reasons To Reject:**

1. The experiments primarily focus on small language models, and as a result, the paper does not effectively demonstrate the impact of the proposed method on large language models like ChatGPT and GPT-4.

2. The proposed strategies for Tree Prompt, including learning, verbalizer, and extensions, lack proper organization in the paper. As a reader, one needs to constantly refer back to previous pages to find relevant information. It would be beneficial to have an overall figure that depicts the training process and the different extensions for Tree Prompting and baselines in a more coherent manner. This would enhance the clarity and understanding of the proposed approach.

**Reproducibility:**

4: Could mostly reproduce the results, but there may be some variation because of sample variance or minor variations in their interpretation of the protocol or method.

**Reviewer Confidence:**

3: Pretty sure, but there's a chance I missed something. Although I have a good feel for this area in general, I did not carefully check the paper's details, e.g., the math, experimental design, or novelty.

---

> ### Author Rebuttal · Authors · 2023-08-29
>
> Thank you for your comments.
>
> ### Re: The experiments primarily focus on small language models, and as a result, the paper does not effectively demonstrate the impact of the proposed method on large language models like ChatGPT and GPT-4.
>
> Our focus on smaller models has been dictated by computational constraints as well as a desire to create solutions that are practical for applications with limited computational resources. That said, we have expanded our experiments to include results on a slightly larger model—Llama-2, with 7 billion parameters. Here are the updated results added to Table 1:
>
> | Method |   SST2 |   SUBJ |   MPQA |   AGNews |   CB |    CR |   DBPedia |    MR |   RTE |   TREC |   Emotion |   FPB |   IMDB |   mean | LLM Calls |
> |-----------------------------------------------------|-------|-------|-------|---------|-----|------|----------|------|------|-------|----------|------|-------|-------|----|
> | FSPrompting |   92.7 |   55.1 |   78.6 |     84.9 | 55.4 |  90.9 |      94.0 |  90.8 |  59.2 |   80.2 | 37.5 |  70.2 |   60.1 |   73.1 | 1.0 |
> | Ensemble |   95.1 |   84.1 | 85.3 | 86.6 | 64.9 |  90.4 | 98.2 |  94.7 |  71.7 |   83.9 | 47.6 |  70.2 |   86.0 |   81.4 | 40.0 |
> | TreePrompt Ensemble |   94.5 | 90.0 | 86.6 | 88.2 | 76.8 |  89.2 | 98.6 |  93.8 |  73.6 |   89.7 | 57.2 |  84.9 | 93.5 |   85.9 | 6.6 |
> | TreePrompt | 93.6 |   89.2 |   85.2 | 88.5 | 75.0 |  88.5 | 98.6 |  93.9 |  69.7 |   88.0 | 54.8 |  84.0 |   93.8 | 84.8 | 35.4 |
>
> We find it encouraging that the performance improves with the quality of the base language model. Specifically, Llama-2 exhibits an average accuracy of 84.8%, as compared to GPT-2 XL's 77.2%. We hope this additional result offers a clearer picture of how TreePrompt scales with the performance of the base language model.
>
> ### Re: Has the author explored training Tree Prompt on smaller language models, such as GPT-2 small, and subsequently evaluating its performance on larger language models? Additionally, did the author manually inspect the trained Tree Prompt and subsequently test it on large language models?
>
> Thanks for this interesting question. To investigate this issue, we conducted an experiment where we trained TreePrompt on smaller models (e.g., GPT-2 small) and subsequently evaluated its performance on larger language models (e.g., GPT-2 medium, large, and even GPT-J). We present the results in the tables below. These tables show the test ROC AUC achieved on an average of four datasets (SST2, Emotion, Financial Phrasebank, and Rotten Tomatoes) when applying various strategies, such as using a single prompt, tree prompting, and gradient-boosted tree prompting.
>
> **Single prompt**
>
> |     model         | GPT-2 (117M) | GPT-2 (345M) | GPT-2 (762M) | GPT-2 (1.5B) | GPT-J (6B) |
> | ----------- | ----------- | ----------- | ----------- | ----------- | --------- |
> | GPT-2 (117M) |         0.68 |         0.63 |         0.69 |     **0.69** |       0.64 |
> | GPT-2 (345M) |         0.65 |         0.69 |         0.71 |     **0.73** |       0.68 |
> | GPT-2 (762M) |         0.64 |         0.64 |     **0.75** |         0.73 |       0.66 |
> | GPT-2 (1.5B) |         0.66 |         0.65 |         0.74 |     **0.74** |       0.68 |
> | GPT-J (6B)   |         0.66 |         0.66 |         0.71 |     **0.72** |        0.7 |
>
> **TreePrompt**
>
> | model   | GPT-2 (117M) | GPT-2 (345M) | GPT-2 (762M) | GPT-2 (1.5B) | GPT-J (6B) |
> | :----------- | -----------: | -----------: | -----------: | -----------: | ---------: |
> | GPT-2 (117M) |     **0.83** |         0.71 |         0.77 |         0.77 |       0.72 |
> | GPT-2 (345M) |         0.75 |     **0.84** |         0.79 |         0.78 |       0.72 |
> | GPT-2 (762M) |         0.73 |         0.72 |     **0.87** |         0.77 |       0.72 |
> | GPT-2 (1.5B) |         0.72 |         0.74 |         0.78 |     **0.87** |       0.72 |
> | GPT-J (6B)   |         0.72 |         0.69 |          0.8 |         0.74 |   **0.85** |
>
> **TreePrompt (GBDT)**
>
> | model   | GPT-2 (117M) | GPT-2 (345M) | GPT-2 (762M) | GPT-2 (1.5B) | GPT-J (6B) |
> | ----------- | ----------- | ----------- | ----------- | ----------- | --------- |
> | GPT-2 (117M) |     **0.85** |         0.73 |          0.8 |         0.79 |       0.76 |
> | GPT-2 (345M) |         0.76 |     **0.86** |          0.8 |         0.77 |       0.74 |
> | GPT-2 (762M) |         0.75 |         0.73 |      **0.9** |         0.81 |       0.78 |
> | GPT-2 (1.5B) |         0.77 |         0.74 |         0.82 |      **0.9** |       0.76 |
> | GPT-J (6B)   |         0.76 |         0.74 |         0.82 |          0.8 |   **0.89** |
>
> It appears that TreePrompt models don't transfer as seamlessly across different sizes of language models as we had hoped. In general, we find that performance is the best when the decision tree is learned on the same language model that is used for inference. This suggests that while TreePrompt boosts performance within a particular model, its knowledge does not easily transfer between models of different sizes (at least for the small scale language models that we tested).
>
> ### Re: In Table 2, does Tree Prompting achieve superior performance on the DBPedia dataset?
>
> Upon reviewing our data, we found an oversight: the performance numbers for MR and DBPedia were inadvertently swapped when we rearranged columns based on dataset sizes. In the corrected table, Tree Prompting achieves a performance of 97.2% on the DBPedia dataset, while the performance on MR should be 86.2%, and both numbers should be bolded. We appreciate your keen attention to detail and have corrected this error in our revised manuscript.

---

### Official Review · Reviewer_8DXL · 2023-08-04

**Soundness:** 4

**Excitement:**

4: Strong: This paper deepens the understanding of some phenomenon or lowers the barriers to an existing research direction.

**Paper Topic And Main Contributions:**

The paper explores prompting language models (LMs) on classification tasks by constructing a decision tree composed of prompts. The proposed Tree prompting method outperforms baseline methods showcased in the paper and is competitive with fine-tuning methods. Additionally, a notable advantage of this approach is the enhanced interpretability of models it offers

**Questions For The Authors:**

A. Can you provide further clarity on the method used to identify prompts for features mentioned in lines 133-143 and $\Theta_1$ - $\Theta_4$ in Figure 1?

**Reasons To Accept:**

- This paper presents a novel Tree prompting method wherein LMs are prompted using a structured decision tree of prompts. A potential advantage of this method is the enhanced interpretability of models.
- Several ways to find prompts for features in the decision tree are proposed.
- Empirical results show that Tree prompting method outperforms the baseline methods presented in the paper.

**Reasons To Reject:**

- While Section 3.1 outlines several ways to construct prompts for features, this paper does not provide performance results for these methods, which leaves readers in the dark regarding which method might yield superior results.
- The main results in Table 1 lack a comprehensive comparison to SOTA methods. I mean, the improvements from the baselines are good but I cannot assess how this pushes the current SOTA. The paper limits its comparison with KNN prompting to GPT2-Large. However, given that the improvements derived from KNN prompting aren't markedly significant and the fact that multiple LMs are evaluated in both Tree prompting and KNN prompting, a broader comparison across different LMs would be more informative.
- The paper is unclear in some technical details. For instance, the ensembling method used in _TreePrompt Ens_ is not illustrated, and the stopping condition of constructing the decision tree in _TreePrompt_ is not elucidated.

**Reproducibility:**

4: Could mostly reproduce the results, but there may be some variation because of sample variance or minor variations in their interpretation of the protocol or method.

**Reviewer Confidence:**

3: Pretty sure, but there's a chance I missed something. Although I have a good feel for this area in general, I did not carefully check the paper's details, e.g., the math, experimental design, or novelty.

**Typos Grammar Style And Presentation Improvements:**

- It appears that the results for the SUBJ and RTE datasets in Table 2 are swapped.

---

> ### Author Rebuttal · Authors · 2023-08-29
>
> Thank you for your comments.
>
> ### Re: While Section 3.1 outlines several ways to construct prompts for features, this paper does not provide performance results for these methods,
>
> Indeed, the primary focus of Table 1 is on "few-shot prompts," which is the main prompt-construction setting discussed at the outset of Section 3. However, results for alternative settings that are covered under Section 3.1 “Extensions” can be found in other tables:
> - "Instruction Prompts" results are presented in Tables 5 and 8.
> - "kNN Prompting Features" results are shown in Table 2.
>
> Regarding "Dynamic Prompts," due to the computational demands involved—each decision tree node's construction requires running the autoprompting method—we opted for a qualitative presentation in Figure 4 to showcase its interpretability.
>
> We acknowledge the need for greater clarity in distinguishing between the main contribution and these extensions in the paper, and will revise the manuscript accordingly.
>
> ### Re: The main results in Table 1 lack a comprehensive comparison to SOTA methods.
>
> We will incorporate SOTA results from a recent preprint [1]. Additionally, we have expanded our evaluations to include the latest 7-billion parameter Llama-2 model, as suggested by reviewer kxdC. The results are presented below:
> | Method |   SST2 |   SUBJ |   MPQA |   AGNews |   CB |    CR |   DBPedia |    MR |   RTE |   TREC |   Emotion |   FPB |   IMDB |   mean | LLM Calls |
> |-----------------------------------------------------|-------|-------|-------|---------|-----|------|----------|------|------|-------|----------|------|-------|-------|------|
> | FSPrompting |   92.7 |   55.1 |   78.6 |     84.9 | 55.4 |  90.9 |      94.0 |  90.8 |  59.2 |   80.2 | 37.5 |  70.2 |   60.1 |   73.1 | 1.0 |
> | Ensemble |   95.1 |   84.1 | 85.3 | 86.6 | 64.9 |  90.4 | 98.2 |  94.7 |  71.7 |   83.9 | 47.6 |  70.2 |   86.0 |   81.4 | 40.0 |
> | TreePrompt | 93.6 |   89.2 |   85.2 | 88.5 | 75.0 |  88.5 | 98.6 |  93.9 |  69.7 |   88.0 | 54.8 |  84.0 |   93.8 | 84.8 | 6.6  |
> | TreePrompt Ensemble |   94.5 | 90.0 | 86.6 | 88.2 | 76.8 |  89.2 | 98.6 |  93.8 |  73.6 |   89.7 | 57.2 |  84.9 | 93.5 |   85.9 | 35.4  |
> | SOTA [1] | 97.4 | - | - | 96.4 | - | - | - | 92.4 | - | - | - | - | - | - | -
>
> For the datasets that overlap with the SOTA, our results are on average 3.2 accuracy points below. However, it's crucial to note that the SOTA model is significantly larger (GPT-3 text-davinci-003, with 175 billion parameters) compared to our Llama-2 model, which has 7 billion parameters. We also observe that better pre-trained models yield better results; for instance, Llama-2's average score of 84.8% significantly outperforms GPT-2 XL’s average of 77.2%. Furthermore, we'd like to emphasize that the prompting techniques used in [1] are complementary to TreePrompt.
>
> ### Re: the ensembling method used in TreePrompt Ens is not illustrated, and the stopping condition of constructing the decision tree in TreePrompt is not elucidated.
>
> We apologize for not clearly specifying the ensembling method and stopping conditions used in TreePrompt. We will add the below information to the manuscript:
> - Ensembling Hyperparameters: We create ensembles of 100 trees, each with max depth 3, and we use a learning rate of 0.1.
> - Stopping Condition: In TreePrompt, the construction of a decision tree stops after reaching 40 splits. This number was chosen to match the number of splits used in our greedy and boosting ensemble baselines.
>
> ### Re: Can you provide further clarity on the method used to identify prompts for features mentioned in lines 133-143 and  -  in Figure 1?
> Thank you for your question. For the segment between lines 133-143 and in Figure 1, which describe our approach to using "few-shot prompts" with k shots, the method works as follows:
>
> We construct a prompt by concatenating k text-label pairs, formatted in the manner of:
> ```
> Input: example_text_1
> Output: example_label_1
> ...
> Input: example_text_k
> Output: example_label_k
> Input: test_sentence
> Output: [logits]
> ```
> The prompt is then processed through the language model to produce logits for the test sentence. A verbalizer is then applied to these logits, transforming them into discrete class names. These class names serve as the features utilized in the decision tree nodes.
>
> We will make sure to elaborate on this method in the revised manuscript for added clarity.
>
> ### References
> [1] Text Classification via Large Language Models (https://arxiv.org/abs/2305.08377)

---

### Official Review · Reviewer_kxdC · 2023-08-04

**Soundness:** 4

**Excitement:**

4: Strong: This paper deepens the understanding of some phenomenon or lowers the barriers to an existing research direction.

**Paper Topic And Main Contributions:**

This paper presents a method for better prompting LMs without fine-tuning. The proposed method, Tree Prompting, works similarly to a decision tree. For each node of the decision tree, generated labels of individual prompts are regarded as the split features. The authors also present several extensions of the proposed method, including human-written instruction prompts, dynamic prompts, $k$NN prompting as split features, and tree ensemble.

Experiments are conducted on 13 text classification datasets. Results show that Tree Prompting outperforms various baselines across multiple LM sizes.

**Questions For The Authors:**

* Question A: Do "dynamic prompts" apply during inference time?
* Question B: As for $k$NN-prompting, is my following understanding correct? At each node, there are a bunch of prompts. The LM does inference for each prompt and the final feature is the NN label.

**Reasons To Accept:**

* The proposed method is novel and interesting. It is a nice combination of traditional simple methods and current LMs.
* Experimental results on multiple datasets indicate the effectiveness of the proposed method.

**Reasons To Reject:**

* It would be better if the authors could provide more insights into the failures of baselines. For example, as shown in Table 1, on TREC, Greedy and Boosting show similar performances with GPT-2 XL, but Boosting significantly underperforms Greedy with GPT-2 Large and Medium.
* The authors did not show the effectiveness of their method using the most recent LLMs, such as LLama. The computational budget should be affordable since the authors have adopted GPT-2 XL and GPT-J. Recent LLMs should be included for comprehensive evaluation.

**Reproducibility:**

3: Could reproduce the results with some difficulty. The settings of parameters are underspecified or subjectively determined; the training/evaluation data are not widely available.

**Reviewer Confidence:**

4: Quite sure. I tried to check the important points carefully. It's unlikely, though conceivable, that I missed something that should affect my ratings.

---

> ### Author Rebuttal · Authors · 2023-08-29
>
> Thank you for your comments.
>
> ### Re: It would be better if the authors could provide more insights into the failures of baselines.
>
> We've conducted further analysis on the training and testing performances, and these analyses will be included in the updated appendix. Below are the key takeaways:
> - FSPrompting: Our data shows that FSPrompting generally underfits, indicated by its low overall performance and minimal difference between training and testing accuracies.
> - Boosting vs. Greedy: The relative performance of Boosting and Greedy becomes more nuanced when considering variations in model size. Larger models serve as more expressive base learners; however, they are also more susceptible to overfitting. The nuances are further complicated by the fact that we use pretrained models, which themselves have varying amounts of pretraining data depending on the model size.
>
> ### Re: The authors did not show the effectiveness of their method using the most recent LLMs, such as LLama.
>
> Thanks for the suggestion. We have now conducted experiments using the Llama-2 model with 7 billion parameters. The updated performance comparisons will be added to Table 1. Our preliminary results indicate that TreePrompt continues to show strong performance when compared with baselines.
>
> Here's a snippet of the additional rows in Table 1 for Llama-2:
>
>
> | Method |   SST2 |   SUBJ |   MPQA |   AGNews |   CB |    CR |   DBPedia |    MR |   RTE |   TREC |   Emotion |   FPB |   IMDB |  Mean | LLM Calls |
> |-----------------------------------------------------|-------|-------|-------|---------|-----|------|----------|------|------|-------|----------|------|-------|-------|----|
> | FSPrompting |   92.7 |   55.1 |   78.6 |     84.9 | 55.4 |  90.9 |      94.0 |  90.8 |  59.2 |   80.2 | 37.5 |  70.2 |   60.1 |   73.1 | 1.0 |
> | Ensemble |   95.1 |   84.1 | 85.3 | 86.6 | 64.9 |  90.4 | 98.2 |  94.7 |  71.7 |   83.9 | 47.6 |  70.2 |   86.0 |   81.4 | 40.0 |
> | TreePrompt Ensemble |   94.5 | 90.0 | 86.6 | 88.2 | 76.8 |  89.2 | 98.6 |  93.8 |  73.6 |   89.7 | 57.2 |  84.9 | 93.5 |   85.9 | 6.6 |
> | TreePrompt | 93.6 |   89.2 |   85.2 | 88.5 | 75.0 |  88.5 | 98.6 |  93.9 |  69.7 |   88.0 | 54.8 |  84.0 |   93.8 | 84.8 | 35.4 |
>
>
> ### Re: Do "dynamic prompts" apply during inference time?
>
> Thank you for your question; we apologize for any confusion. To clarify: the "dynamic prompts" are generated during the training phase, where they are optimized to suit each specific node in the decision tree. In other words, the prompts are "dynamic" in that they are adapted to the specific characteristics of the subset of data that flows to each node during training.
>
> However, once the model is trained, the decision tree—including its structure and associated prompts—is fixed. So while the prompts are dynamically generated during training, they are static during inference. We'll make sure to include this clarification in the updated paper.
>
> ### Re: As for kNN-prompting, is my following understanding correct? At each node, there are a bunch of prompts. The LM does inference for each prompt and the final feature is the NN label.
>
> Your understanding is correct. At each node in the decision tree, we use the language model to generate logits for both the test example and a set of training examples. We then find the nearest neighbor in the set of training examples based on the generated logits. The label of this nearest neighbor is used as the final feature for that node.

---

### Meta-Review · Area_Chair_KKuG · 2023-09-11

**Recommendation:** 4

**Metareview:**

This paper introduces an innovative approach called Tree Prompting, which aims to enhance the performance of Language Models (LMs) without the need for fine-tuning. In this method, Tree Prompting operates in a manner akin to a decision tree. Each node of the decision tree is associated with generated labels derived from individual prompts, serving as the split features. Furthermore, the authors present various extensions of this novel method, including the incorporation of human-written instruction prompts, dynamic prompts, the use of Neural Network (NN) prompting as split features, and the implementation of a tree ensemble technique. However, it would be better if the authors could provide more insights into the failures of baselines.

---

### Decision · Program_Chairs · 2023-10-07

**Decision:**

Accept-Main

**Comment:**

This paper introduces an innovative approach called Tree Prompting, which aims to enhance the performance of Language Models (LMs) without the need for fine-tuning. In this method, Tree Prompting operates in a manner akin to a decision tree. Each node of the decision tree is associated with generated labels derived from individual prompts, serving as the split features. Furthermore, the authors present various extensions of this novel method, including the incorporation of human-written instruction prompts, dynamic prompts, the use of Neural Network (NN) prompting as split features, and the implementation of a tree ensemble technique. However, it would be better if the authors could provide more insights into the failures of baselines.